# Cefiderocol Efficacy in a Real-Life Setting: Single-Centre Retrospective Study

**DOI:** 10.3390/antibiotics12040746

**Published:** 2023-04-13

**Authors:** Gabriele Palermo, Alice Annalisa Medaglia, Luca Pipitò, Raffaella Rubino, Manuela Costantini, Salvatore Accomando, Giovanni Maurizio Giammanco, Antonio Cascio

**Affiliations:** 1Department of Health Promotion, Mother and Child Care, Internal Medicine and Medical Specialties “G D’Alessandro”, University of Palermo, 90127 Palermo, Italy; gabrielepalermo01@gmail.com (G.P.); lucapipito@gmail.com (L.P.); salvatore.accomando@unipa.it (S.A.); giovanni.giammanco@unipa.it (G.M.G.); 2Infectious and Tropical Disease Unit and Sicilian Regional Reference Center for the Fight against AIDS, AOU Policlinico “P. Giaccone”, 90127 Palermo, Italy; alicemedaglia@gmail.com (A.A.M.); raffaella.rubino@policlinico.pa.it (R.R.); 3Antimicrobial Stewardship Team, AOU Policlinico “P. Giaccone”, 90127 Palermo, Italy; 4UOC Farmacia, AOU Policlinico “P. Giaccone”, 90127 Palermo, Italy; manuela.costantini@policlinico.pa.it; 5Microbiology and Virology Unit, Department of Health Promotion, Mother and Child Care, Internal Medicine and Medical Specialties “G D’Alessandro”, University of Palermo, 90127 Palermo, Italy

**Keywords:** cefiderocol, carbapenem-resistant, CRAB

## Abstract

The current carbapenem-resistant gram-negative bacteria (CR-GN) treatment guidelines lack strong evidence about cefiderocol (CFD) efficacy against CR-GN, especially CRAB. The study’s purpose is to evaluate the effectiveness of CFD in a real-life setting. We made a single-center retrospective study of 41 patients who received CFD in our hospital for several CR-GN infections. Bloodstream infections (BSI) affected 43.9% (18/41) of patients, while CRAB affected 75.6% (31/41) of isolated CR-GN patients. Thirty-days (30-D) all-causes mortality affected 36.6% (15/41) of patients, while end-of-treatment (EOT) clinical cure affected 56.1% (23/41). Finally, microbiological eradication at EOT affected 56.1% (23/41) of patients. Univariate and multivariate analysis showed that septic shock is an independent factor associated with mortality. Subgroup analyses showed no difference in CFD effectiveness between monotherapy and combination therapy.

## 1. Introduction

Hospital-acquired infections (HAIs) are common complications seen in hospitalized patients that increase mortality, costs, and length of stay [1]. Carbapenem-resistant gram-negative bacteria (CR-GN) are the principal cause of HAIs and represent a great challenge for public health professionals [1]. Previously, the only available treatments for CR-GN infections were associated with several adverse effects, poor efficacy, and the development of antibiotic resistance [1].

Today, safer and more effective drugs active against CR-GN are available [2,3,4], among which cefiderocol (CFD), a first-in-class siderophore cephalosporin, has a broader spectrum of action that also includes the Ambler class B carbapenemases and carbapenem-resistant lactose non-fermenters *Acinetobacter baumannii* (CRAB), *Pseudomonas aeruginosa* (CRPsA), and *Stenotrophomonas maltophilia* [5].

Several expert groups tried to create pathogen-focused or source-focused therapeutic algorithms, choosing the right place in the therapy to use these new drugs based on their pharmacokinetic/pharmacodynamic (PK/PD) features [6,7]. Furthermore, international and Italian guidelines on CR-GN infection treatment were published in 2022 [8,9,10]. All guidelines highlight the lack of strong evidence about CFD efficacy against CR-GN, especially against CRAB (particularly after the controversial results of the CREDIBLE-CR study were published [11]). For this reason, CFD was not recommended as a first-line treatment against CR-GN [8,9,10].

Real-world data shows a large variability of CFD efficacy in different infections [12,13,14,15,16,17,18,19,20]. Falcone et al. found that although there was a significant benefit to clinical outcomes in BSIs of CFD over colistin-containing regimens (*p* = 0.007), this benefit was not found in VAPs caused by CRAB (*p* = 0.918); the authors supposed that these findings could be due to baseline features of these patients, such as COVID-19 coinfection, which makes it difficult to assess the actual efficacy of antibiotic therapy [12]. 

This paper aims to increase the available evidence about the efficacy of CFD in a real-life setting.

## 2. Materials and Methods

We conducted a retrospective study on all patients admitted to “AOU Policlinico P. Giaccone” of Palermo hospital between September 2021 and July 2022 and treated with CFD. Patients were identified by CFD prescriptions obtained from hospital pharmacy department. All patients’ data were obtained from the intranet hospital system and standardized discharge forms. Data were then anonymized and entered into a specific database. Continuous variables were expressed in mean and standard deviation (SD), median and interquartile range (IQR) terms. They were compared with the Student t-test or the Mann–Whitney U test as appropriate. Categorical variables were expressed in numbers of events and their percentage, while the χ^2^ test was performed to compare them. Two-tailed tests were used to determine statistical significance; we considered significant a *p* value of <0.05. Multivariate logistic regression analysis was performed to identify independent factors related to 30-D mortality rates. We included in the multivariate model all variables with a *p* value <0.05 that emerged from univariate analysis. Subgroup analyses were performed on patients with CRAB infections and patients treated in monotherapy or combination therapy. Furthermore, subgroup analysis of each type of infection was performed to see their features and impact on patients’ outcomes.

### 2.1. Patients and Infections Profiles

The Charlson comorbidity index (CCI) [21], previous (12 months) hospital admissions, (3 months) bacterial infections and antibiotic treatments (during in-hospital stay) before CFD administration were considered. Infection types were defined according to sources of collected bacterial isolates and clinical judgment. For each patient, there could be more than one source of infection (for example, simultaneous pulmonary and urinary tract involvement). They were singularly considered in statistical analysis. Infections were defined as bloodstream infections if there was documented positive blood culture. We judged the presence of septic shock at infection onset as a severity illness factor and as infection onset the same day CFD was started. Infections were defined as hospital-acquired if bacterial isolates were collected 48 h after admission to the hospital.

### 2.2. Microbiology

Bacterial isolate identification was performed by Matrix Assisted Laser Desorption Ionization Time of Flight Mass Spectrometry (MALDI-ToF MS) (MALDI Biotyper, Bruker Daltonics, Germany). Carbapenems susceptibility tests were performed using automized systems. Minimum inhibitory concentrations (MICs) were classified according to breakpoints established by the European Committee on Antimicrobial Susceptibility Testing [22]. CFD MIC was tested with disk diffusion method in Mueller–Hinton agar. When it was isolated as more than one bacterium in the same patient, each isolated bacterium was singularly considered in statistical analysis.

### 2.3. Therapy Variables

CFD was administered as a 3-h standard infusion of 2 g intravenously every 8 h, with adjustments for renal impairment according to manufacturer recommendations. We described any adverse events during antibiotic treatment. CFD was used in either monotherapy or in combination with another active drug; combination therapy regimens included fosfomycin, at the dose of 6 g intravenously every 8 h or 4 g every 8 h, or colistin, at the loading dose of 9.000.000 UI of colistimethate sodium followed by 4.500.000 UI every 12 h as a maintaining dose. We described the length of therapeutic regimens and length of stay in the hospital between admission and the start of CFD in the total population and each analyzed subgroup.

### 2.4. Outcomes

The primary outcome was 30-D all-cause mortality, defined as the occurrence of death within 30 days from the start of treatment with CFD. Secondary outcomes were clinical response during the first 72 h of therapy, defined as an initial clinical response in which it was not necessary to switch therapy or the patient did not die; clinical cure at end-of-therapy (EOT), defined as symptoms resolution and clearance of inflammatory markers; need to switch therapy, defined as interruption of CFD and switch therapy due to clinical or microbiological failure; and microbiological eradication, defined as documented or presumed eradication of isolated bacteria at EOT.

## 3. Results

### 3.1. Study Population

#### 3.1.1. Demographic and Anamnestic Features

Data from 41 patients and 62 infections were analyzed. Their characteristics are shown in Table 1. The distribution for sex and age classes is shown in Figure 1.

#### 3.1.2. Infection Features

All infections were hospital-acquired and were diagnosed in medical wards (48.8%), in surgical wards (19.5%), or in intensive-care units (31.7%). Among all patients, 34.2% were in septic shock at infection onset. In total, 43.9% of patients had BSIs and 41.5% had hospital-acquired pneumonia (HAP), of which cases 94.1% were ventilator-associated (VAP). All infection features are summarized in Table 1 and shown in Figure 2.

#### 3.1.3. Microbiological Features

All bacterial isolates were carbapenem-resistant (MIC > 8 mg/L) [15]; they are summarized in Table 1 and shown in Figure 3. The most isolated bacteria were CRAB (75.6%). In only two cases of CFD MIC were tested: 1 VAP case was caused by CRAB (MIC = 0.38 μg/mL) and 1 cUTI case was caused by KPC NDM (MIC = 1 μg/mL)

#### 3.1.4. Treatment Features

Treatment features are summarized in Table 2. CFD was started after a median of 21 days (IQR, 5–25) from hospital admission. It was administered in monotherapy in 75.6% of cases. In the other cases, it was administered with fosfomycin (14.6%, 6/41) or colistin (9.6%, 4/41). The median duration of therapy was 9 days (IQR, 6–19 days). Adverse events occurred in 4.9% (2/41) of cases; hypersensitivity drug reactions (maculopapular rash) occurred after the first 48 h of treatment and bronchospasm occurred after the first dose.

#### 3.1.5. Outcomes

Outcomes are summarized in Table 2. Overall, in-hospital mortality was 46.3%, while 30-D all-cause mortality was 36.6%. Clinical response during the first 72 h of treatment was 80.5%., while clinical cure at EOT was 56.1%. Microbiological eradication at EOT was 80.5%. In three cases (7.3%), a switch to ceftazidime/avibactam was undertaken: two switches stemmed from the above-mentioned adverse drug reaction and one from CRKP-KPC targeted therapy.

### 3.2. Subgroup Analyses

#### 3.2.1. Monotherapy vs. Combination Therapy

Patients’ infections’ microbiological and treatment features and outcomes for both subgroups are summarized in Table 1 and Table 2 and Figure 2 and Figure 3. No statistical differences in terms of demographic and clinical characteristics were observed between them (see Table 1). In the combination therapy subgroup, the percentage of CRPsA was statistically significantly higher (*p* = 0.047). The median duration of CFD therapy was 9 days (IQR, 5–16 days) when used in monotherapy and 13 days (IQR, 6–21 days) days when used in combination therapy (*p* = 0.36). In the monotherapy subgroup, although there was a lower 72 h clinical response (77.4% vs. 90.0%), we documented lower 30-D mortality (35.5% vs. 40.0%) and higher clinical cure and microbiological eradication at EOT (respectively 58.1% vs. 50.0% and 87.1% vs. 60.0%), even if none of these differences was statistically significant (see Table 2).

#### 3.2.2. CRAB Infections

CRAB infections were 75.6% (31/41) of the total and are summarized in Table 1 and Table 2. CFD was used in this setting mainly as monotherapy (80.6%). In the remaining cases, it was associated with fosfomycin (12.9%) or colistin (6.40%). In this subgroup, 41.6% of patients had HAP (all VAPs) and 48.4% had a BSI. Overall, 30-D mortality was 35.5%, while between VAPs was 61.5% and between BSIs infections was 46.7%. Clinical response during the first 72 h of treatment was 90.3%. The clinical cure at EOT was 64.5%. Microbiological eradication at EOT was 80.6%. Under no circumstance was it necessary to switch CFD with another therapeutic regimen.

#### 3.2.3. 30-D Survivors vs. Non-Survivors

The characteristics of patients, their infections, antibiotic treatment and outcomes of both subgroups are summarized in Table 1 and Table 2 and Figure 2 and Figure 3. No statistically significant difference was observed in the sex of patients (*p* = 0.27). Patients who died were older than patients who survived (with a median age of 70 years (IQR, 70–78 years) and 60 (IQR, 51–75 years), respectively) and had more comorbidities (with a CCI ≥ 4 of 93.3% and 57.7%, respectively); these differences were statistically significant at the univariate analysis (*p* = 0.028 and *p* = 0.01, respectively), though they were not confirmed in multivariate analysis. Hospital wards at infection onset were similar in both subgroups.

Regarding infection variables, we observed a higher percentage of CNS infections, HAPs and cIAIs among patients who died and a higher percentage of cUTIs and SSTIs in survivors. However, these differences were not statistically significant. A higher septic shock percentage was observed among patients who died; this difference was statistically significant (*p* = 0.0008; COR 11 [95% CI, 2.42–49.91]) and persisted in the multivariate model (*p* = 0.02; AOR 8.1 [95% CI, 1.35–48.46]).

No statistically significant difference was observed between the two subgroups regarding microbiological and treatment variables. Figures for microbiological eradication at EOT and clinical response after 72 h of treatment were similar in both subgroups, as was the percentage of combination and monotherapy in the 2 subgroups.

#### 3.2.4. Types of Infections

In Table 3, we described each type of infection by considering their microbiological and therapeutical features and their impact on patients’ outcomes.

## 4. Discussion

In our real-life retrospective study, 30-D mortality was 36.6% and clinical cure at EOT was 56.1%. Population baseline features may have influenced both results. Indeed, as expected, age and CCI were higher among patients that expired. Subgroup analysis based on site of infections (Table 3) shows the heterogeneity of infections included in this study and permits comparison of results with further studies and meta-analysis of each type of infection. We wanted to compare our results with those of other similar studies. The main real-life studies regarding CFD are reported in Table 4. The 30-D mortality in these studies ranged from 12.5 to 60% [14,15,16,17,18,19,20,21,22]. This wide variability is probably due to the different conditions of the patients described in these studies. The 30-D mortality in our study is similar to the study described by Falcone et al. (36.6% vs. 34%), who used a similar sample size of patients (41% vs. 47%); however, 30-D mortality was notably lower compared to the study described by Pascale et al. (36.6% vs. 55%), where all patients were affected by critical COVID-19, necessitating mechanical ventilation.

An interesting finding in our study was the different distribution of types of infections among 30-D survivors and non-survivors. HAPs, CNS infections and cIAIs were more represented in 30-D non-survivors.

In our study, patients with HAPs/VAPs represented 41.5% of total infections and 15/17 patients treated with CFD alone. We observed high mortality (52.9%) and low clinical cure (29.4%) in these patients; however, we also observed concomitant high microbiological eradication (88.2%), allowing us to assume that other clinical conditions beyond bacterial infection determined the outcomes of these patients’ treatment. HAPs are caused mainly by CR-GN and had high mortality despite any medical intervention [23,24,25,26], PK/PD data showed that CFD is a valid option for treating these infections [27]. Real-world evidence shows a 30-D mortality ranging from 30.8% to 80% according to the pathogen, respiratory-impairing or COVID-19 coinfection (Table 4). 

In our study, two patients with CNS CRAB infections were treated and both died. Although evidence on CNS infections CFD use is lacking (Table 4), few PK/PD studies suggest that CFD could be a promising option for lactose non-fermenters CR-GN treatment, with drug concentration in cerebrospinal fluid above bacterial MIC [17,28]. 

In our study, almost one infection in five patients were cIAI (19.5%, 8/41) and they were represented mainly by walled-off pancreatic infected necrosis, conditions with high mortality in which surgical source control is difficult [29]. Despite this problematic setting, 5/8 cases were considered microbiologically eradicated. Meschiari et al. described 4 cIAIs (3 peritonitis and 1 cholangitis) sustained by CRPsA where 30-D mortality was 50% but 100% of infections were considered microbiological cured.

Multivariate analysis of our population showed that septic shock was the only independent factor associated with 30-D mortality. Our study confirms the real-life efficacy of CFD in patients with septic shock, having a high microbiological eradication (85.7%). The above is likely achieved thanks to the PK/PD CFD characteristics in these challenging clinical settings [19].

In our opinion, the most interesting finding of this study was that there was no statistical difference in the clinical and microbiological efficacy of CFD when used in monotherapy or combination therapy regimens. The two subgroups were quite similar except for the number of CRPsA isolates, which was higher in the combination therapy subgroup (*p* = 0.47). Noteworthy lower clinical cure and microbiological eradication resulted in the combination therapy subgroup, despite better initial clinical response, which could be explained by infection-related factors (such as a higher percentage of polymicrobial isolates, CRPsA isolates, and cIAIs in the combination therapy subgroup) or other underlying clinical conditions.

The use of CFD in combination therapy is debated in the literature, especially in regard to CRAB infections. The IDSA panel suggests prescribing CFD as a component of a combination regimen until more favorable clinical data on CFD efficacy as a monotherapy are available [10]. Italian societies’ guidelines request further studies to consolidate recommendations on CFD use and evaluate the use of this drug as monotherapy or in a combination therapy regimen with other antibiotics [8]. The rationale of CFD use in combination therapy regimens is based on in vitro data on the potential benefit of adding a different class antimicrobial agent to overcome the non-susceptibility of CR-GN to CFD [30,31,32]. Furthermore, keeping in mind the difficulties of obtaining reliable CFD-susceptibility tests, as highlighted by Eucast [33], recent studies suggest the possibility of underestimating heteroresistant subpopulations and in vivo developed resistance due to inoculum effect [34,35]; this outlook assumes the use of combination therapy regimens in infections at high risk of suboptimal antibiotic exposure, such as cIAIs, osteomyelitis and particular cases of difficult-to-treat VAPs. Real-world clinical data suggest that CFD plus fosfomycin, colistin or tigecycline could be effective against difficult-to-treat *P. aeruginosa* and CRAB infections [8,36,37]. 

To the best of our knowledge, this is the most extensive study comparing the efficacy of CFD as monotherapy and combination therapy. Recently Corcione et al. (Table 4) analyzed CFD efficacy in infections sustained by CR-GN (CRAB 83.3%), finding there was no statistically significant difference in clinical and microbiological outcomes between monotherapy and combination therapy regimens (mainly colistin-based) [16].

Over two-thirds of the bacterial isolates were CRAB and an analysis of this subgroup allows us to make some considerations. In our study, the 31 CRAB infections were mainly represented by VAPs (41.9%) and BSIs (48.4%), with a 30-D mortality of 61.5% and 46.7%, respectively (see Table 1, Table 2 and Table 4). Recently a nationwide study in Italy was published describing a 43.2% 30-D mortality in patients with CRAB BSIs [38], similar to that observed in our study. Two retrospective studies were recently published evaluating CFD efficacy in CRAB infections: Falcone et al. demonstrated that CFD-containing regimens were non-inferior to colistin-based regimens in all infections and superior in BSIs [12]; and Pascale et al. evaluated CFD efficacy in a very difficult setting, such as patients in ICU for critical COVID-19, where CFD as monotherapy demonstrated non-inferiority to BAT [13]. Our mortality rate was similar to the one reported by Falcone et al. (35.5% vs. 34%) but lower in comparison to the one documented by Pascale et al., which is likely higher because of critical features at baseline in that population [12,13].

Two large RCTs analyzed CFD efficacy in CR-GN infections: APEKS-NP (vs. high-dose meropenem) and CREDIBLE-CR (vs. BAT). Although in APEKS-NP trial CFD showed non-inferiority on the treatment of HAPs [39], several debates emerged after the publication of results of the CREDIBLE-CR study [11]. Subgroup analysis in this RCT showed a higher overall mortality in CFD arm between HAPs (42% vs. 18%), BSIs (37% vs. 18%) and CRAB infections (49% vs. 18%). Authors suggest that a poorly balanced distribution of septic shock and admission in ICU variables may have influenced these results. In our study, 30-D mortality, clinical cure and microbiological eradication at EOT created results more similar to real-world studies than those emerged in RCTs (Table 2 and Table 4). This finding can be explained by the different nature of the studies and inclusion criteria.

## 5. Conclusions

In conclusion, our study shows that CFD is a viable therapy option for a challenging group of CR-GN infections, particularly CRAB infections. This medication permitted clinicians to utilize antibiotic regimens in CRAB infections for more days while maintaining the renal function of older patients. According to our data, CFD constitutes a valid antibiotic regimen against CRAB in colistin-sparing and difficult-to-treat infections. This study adds further evidence to support the use of CFD against CR-GN and in monotherapy or combination therapy regimens; this proposal is in spite of this treatment’s various limitations, such as retrospective nature. It is hoped that future RCTs will better establish the correct role for CFD therapy in treating hospital-acquired infections.

## Figures and Tables

**Figure 1 antibiotics-12-00746-f001:**
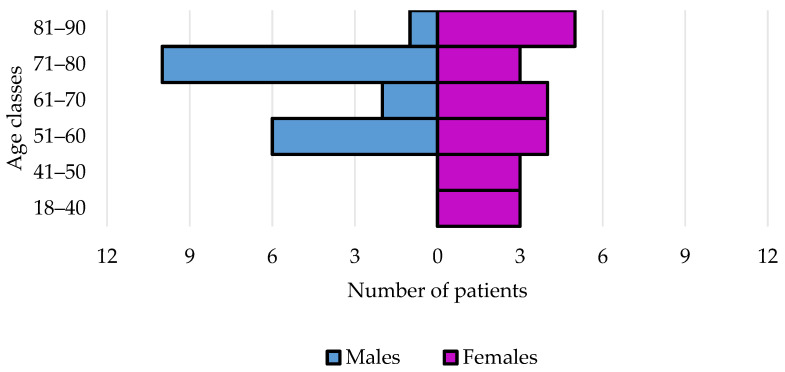
Age and sex of study population that received CFD therapy in our hospital.

**Figure 2 antibiotics-12-00746-f002:**
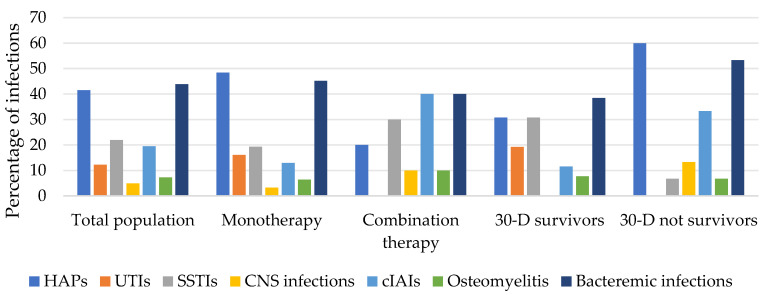
Types of infections of the study population that received CFD therapy in our hospital.

**Figure 3 antibiotics-12-00746-f003:**
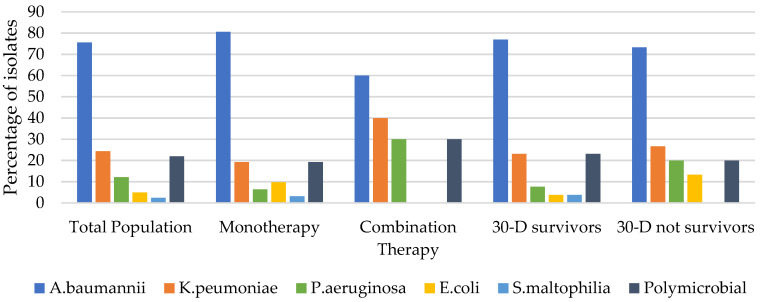
Microbiological isolates of the study population that received CFD therapy in our hospital.

**Table 1 antibiotics-12-00746-t001:** Demographic, clinical and microbiological features of patients who received CFD and subgroup analyses of those who received it as monotherapy and combination therapy regimens, 30-day survivors and non-survivors and infections sustained by carbapenem-resistant *Acinetobacter baumannii*.

Variables	Study Population	Monotherapy	Combination	*p* Value	30-D Survivors	30-D Non-Survivors	*p* Value	CRAB
Patients	n = 41	n = 31	n = 10		n = 26	n = 15		n = 31
Males	20 (48.8)	15 (48.4)	5 (50.0)	0.93	11 (42.3)	9 (60.0)	0.27	16 (51.6)
Females	21 (51.2)	16 (51.6)	5 (50.0)	0.92	15 (57.7)	6 (40.0)	0.27	15 (48.4)
M:F ratio	0.9:1	0.9:1	1:1		0.7:1	1.5:1		1.1:1
Age, median years (IQR)	70 (54–75)	61 (52–76)	72 (64–74)	0.23	60 (51–75)	73 (70–78)	**0.02**	61 (52–73)
Age, years > 70	19 (46.3)	12 (38.7)	7 (70.0)	0.08	8 (30.8)	11 (73.3)	**0.008**	11 (35.5)
CCI, median (IQR)	5 (3–6)	5 (2–6)	4 (3–5)	0.92	4 (2–6)	5 (5–6)	0.07	4 (12.9)
CCI ≥ 4	29 (70.7)	22 (71.0)	7 (70.0)	0.95	15 (57.7)	14 (93.3)	**0.01**	20 (64.5)
Previous in hospital admission *	20 (48.8)	14 (45.2)	6 (60.0)	0.41	11 (42.3)	9 (60.0)	0.27	13 (41.9)
Previous bacterial infections *	17 (41.5)	11 (35.5)	6 (60.0)	0.17	8 (30.8)	9 (60.0)	0.60	12 (38.7)
Previous antibiotic therapy *	34 (82.9)	25 (80.6)	9 (90.0)	0.9	21 (80.7)	12 (80.0)	0.70	25 (80.6)
**Unit at infection onset**								
Medical ward	20 (48.8)	16 (51.6)	4 (40.0)	0.52	13 (50.0)	7 (46.7)	0.83	14 (45.2)
Surgical ward	8 (19.5)	4 (12.9)	4 (40.0)	0.06	6 (23.1)	2 (13.3)	0.44	6 (19.3)
Intensive care unit	13 (31.7)	11 (35.5)	2 (20.0)	0.36	7 (26.9)	6 (40.0)	0.39	11 (35.5)
**Types of infections**								
HAPs	17 (41.5)	15 (48.4)	2 (20.0)	0.11	8 (30.8)	9 (60.0)	0.05	13 (41.9)
cUTIs	5 (12.2)	5 (16.1)	0 (0)	0.17	5 (19.2)	0 (0)	0.06	3 (9.7)
cIAIs	8 (19.5)	4 (12.9)	4 (40.0)	0.06	3 (11.5)	5 (33.3)	0.08	6 (19.3)
SSTIs	9 (21.9)	6 (19.3)	3 (30.0)	0.47	8 (30.8)	1 (6.7)	0.07	8 (25.8)
Osteomyelitis	3 (7.3)	2 (6.4)	1 (10.0)	0.71	2 (7.7)	1 (6.7)	0.90	2 (6.4)
CNS infections	2 (4.9)	1 (3.2)	1 (10.0)	0.39	0 (0)	2 (13.3)	0.05	2 (6.4)
Bloodstream infections	18 (43.9)	14 (45.2)	4 (40.0)	0.77	10 (38.5)	8 (53.3)	0.35	15 (48.4)
Septic shock	14 (34.2)	11 (35.5)	3 (30.0)	0.75	4 (15.4)	10 (66.7)	**0.0008**	10 (32.2)
**Bacterial isolates**								
*Acinetobacter baumannii*	31 (75.6)	25 (80.6)	6 (60.0)	0.18	20 (76.9)	11 (73.3)	0.79	
*Klebsiella pneumoniae*	10 (24.4)	6 (19.3)	4 (40.0)	0.18	6 (23.1)	4 (26.7)	0.79	
*Pseudomonas aeruginosa*	5 (12.2)	2 (6.4)	3 (30.0)	**0.047**	2 (7.7)	3 (20.0)	0.41	
*Escherichia coli*	3 (7.3)	3 (9.7)	0 (0)	0.31	1 (3.8)	2 (13.3)	0.26	
*Stenotrophomonas maltophilia*	1 (2.4)	1 (3.2)	0 (0)	0.56	1 (3.8)	0 (0)	0.44	
Polymicrobial isolates	9 (22.0)	6 (19.3)	3 (30.0)	0.47	6(23.1)	3 (20.0)	0.47	
No isolates	1 (2.4)	1 (3.2)	0 (0)	0.31	1 (3.8)	0 (0)	0.44	

Where not otherwise defined, results are presented as the number of patients and their (%) in total. Here **30-D** = thirty-day; **CRAB** = carbapenem-resistant *Acinetobacter baumannii*; **M:F** = male-to-female ratio; **IQR** = interquartile range; **CCI** = Charlson comorbidity index; **HAP** = hospital-acquired pneumonia; **cIAI** = complicated intrabdominal infections; **cUTI** = complicated urinary-tract infections; **SSTI** = soft-skin tissues infections; **CNS** = central nervous system; ***** see text; in **bold** significant *p* value (< 0.05).

**Table 2 antibiotics-12-00746-t002:** Therapeutic features and outcomes for patients who received CFD. Subgroup analyses of those who received it as monotherapy and combination therapy regimens, 30-day survivors and non-survivors and infections sustained by carbapenem-resistant *Acinetobacter baumannii*.

Variables	Study Population	Monotherapy	Combination	*p* Value	30-D Survivors	30-D Non-Survivors	*p* Value	CRAB
Therapy	n = 41	n = 31	n = 10		n = 26	n = 15		n = 31
Days before therapy, median (IQR) *	21 (14–32)	19 (13–31)	22 (15–33)	0.53	23 (12–32)	17 (14–26)	0.31	21 (14–34)
Days of therapy, median (IQR)	9 (6–19)	9 (5–16)	13 (6–21)	0.36	10 (7–22)	7 (3–14)	0.16	9 (7–21)
Duration of therapy, >9 days	19 (46.3)	13 (41.9)	6 (60.0)	0.31	14 (53.8)	5 (33.3)	0.20	15 (48.4)
Monotherapy regimen	31 (75.6)				20 (76.9)	11 (73.3)	0.79	25 (80.6)
Combination therapy regimen	10 (24.4)				6 (23.1)	4 (26.7)	0.79	6 (19.3)
**Outcomes**								
Microbiological eradication at EOT	33 (80.5)	27 (87.1)	6 (60.0)	0.06	22 (84.6)	11 (73.3)	0.43	25 (80.6)
Clinical cure at EOT	23 (56.1)	18 (58.1)	5 (50.0)	0.65				20 (64.5)
Clinical response in first 72 h *	33 (80.5)	24 (77.4)	9 (90.0)	0.38	22 (84.6)	11 (73.3)	0.43	28 (90.3)
Need to switch *	3 (7.3)	2 (6.4)	1 (10.0)	0.71	2 (7.7)	1 (6.7)	0.90	0 (0)
All-causes hospital mortality	19 (46.3)	14 (45.2)	5 (50.0)	0.78				15 (48.4)
30-D all-causes mortality	15 (36.6)	11 (35.5)	4 (40.0)	0.79				11 (35.5)

Where not otherwise defined, results are presented as the number of patients and their (%) on total. Here **30-D** = thirty-day; **CRAB** = carbapenem-resistant *Acinetobacter baumannii*; **IQR** = interquartile range; **EOT** = end of therapy; ***** see text; in **bold** significant *p* value (<0.05).

**Table 3 antibiotics-12-00746-t003:** Microbiological and therapeutical features and outcomes of each type of infection treated with CFD.

Variables	HAPs	cUTIs	cIAIs	SSTIs	Osteomyelitis	CNS Infections	BSIs
Patients	n = 17	n = 5	n = 8	n = 9	n = 3	n = 2	n = 18
**Bacterial isolates**							
*Acinetobacter baumannii*	13 (76.5)	3 (60.0)	6 (75.0)	8 (88.9)	2 (66.7)	2 (100)	15 (83.3)
*Klebsiella pneumoniae*	2 (11.8)	2 (40.0)	5 (62.5)	3 (33.3)	0 (0)	0 (0)	5 (27.8)
*Pseudomonas aeruginosa*	2 (11.8)	0 (0)	0 (0)	2 (22.2)	1 (33.3)	0 (0)	2 (11.1)
*Escherichia coli*	2 (11.8)	1 (20.0)	0 (0)	0 (0)	0 (0)	0 (0)	1 (5.6)
*Stenotrophomonas maltophilia*	0 (0)	0 (0)	0 (0)	0 (0)	0 (0)	0 (0)	1 (5.6)
Polymicrobial isolates	3 (17.6)	1 (20.0)	3 (37.5)	3 (33.3)	0 (0)	0 (0)	0 (0)
No isolates	1 (5.8)	0 (0)	0 (0)	0 (0)	0 (0)	0 (0)	0 (0)
**Therapy**							
Days of therapy, median (IQR)	8 (2–14)	10 (7–10)	10 (4–24)	11 (8–22)	32 (24–37)	12 (8–16)	9 (7–18)
Monotherapy regimen	15 (88.2)	5 (100)	4 (50.0)	6 (66.7)	2 (66.7)	1 (50.0)	14 (77.8)
Combination therapy regimen	2 (11.8)	0 (0)	4 (50.0)	3 (33.3)	1 (33.3)	1 (50.0)	4 (22.2)
**Outcomes**							
Microbiological eradication at EOT	15 (88.2)	5 (100)	5 (62.5)	8 (88.9)	2 (66.7)	1 (50.0)	14 (77.8)
Clinical cure at EOT	5 (29.4)	4 (80.0)	3 (37.5)	8 (88.9)	2 (66.7)	1 (50.0)	9 (50.0)
Clinical response in first 72 h *	11 (64.7)	4 (80.0)	6 (75.0)	8 (88.9)	3 (100)	2 (100)	16 (88.9)
30-D all-causes mortality	9 (52.9)	0 (0)	5 (62.5)	1 (11.1)	1 (33.3)	2 (100)	8 (44.4)

Where not otherwise defined, results are presented as the number of patients and their (%) on total. **HAP** = hospital-acquired pneumonia; **cIAI** = complicated intrabdominal infections; **cUTI** = complicated urinary-tract infections; **SSTI** = soft-skin tissues infections; **CNS** = central nervous system; **BSI** = bloodstream infection; **IQR** = interquartile range; **EOT** = end of therapy; ***** see text; **30-D** = thirty-days.

**Table 4 antibiotics-12-00746-t004:** Summary of real-life clinical studies on CFD efficacy present in literature and 30-D mortality rates for each type of infection.

Variables	Our Study	Falcone et al. [12]	Pascale et al. [13]	Hoellinger et al. [14]	Weber et al. [15]	Corcione et al. [16]	Meschiari et al. [17]	Bavaro et al. [18]	König et al. [19]	Gavaghan et al. [20]
No. of patients	41	47	42	10	8	18	17	13	5	24
Median age (IQR)	70 (54–75)	63 (53–75)	64 (55–73)	66 (56–71.5)	64 (39–80)	54.5 (35–65)	64 (58–73)	63 (53–69)	55 (41–76)	66.5 (60–74)
HAPs/VAPs	9/17 (52.9)	7/12 (58.3)	NA/14	4/5 (80)	0/2 (0)	4/13 (30.8)	3/8 (37.5)	1/3 (33.3)	1/4 (25)	9/19 (47.4)
CRAB	8/13 (61.5)	7/12 (58.3)	NA/14	1/1 (100)	0	4/13 (30.8)	1/1 (100)	1/2 (50)	1/2 (50)	5/12 (41.7)
*K. pneumoniae*	1/2 (50)	NA	0	0	0	0	1/1 (100)	0	0	2/3 (66.7)
*P. aeruginosa*	0/2 (0)	NA	0	1/2 (50)	0/2 (0)	0	3/8 (37.5)	0/1 (0)	0/2 (0)	3/7 (42.8)
Polymicrobial	2/3 (66.7)	NA	NA	0	0	0	2/3 (66.7)	0	0	2/6 (33.3)
cIAIs	5/8 (62.5)	NA/1	NA	0/1 (0)	1/1 (100)	0	2/4 (50)	0/2 (0)	0	0
cUTIs	0/5 (0)	0	NA	2/2 (100)	0/4 (0)	0	0	0	0	0/1 (0)
SSTIs	1/9 (11.1)	NA/5	NA	0	0	0/1 (0)	0/1 (0)	0/1 (0)	0	2/4 (50)
Osteomyelitis	1/3 (33.3)	NA/0	NA	0	0/1 (0)	0	1/1 (100)	0	0	0
CNS infections	2/2 (100)	NA/1	NA	1/1 (100)	0	0	0/1 (0)	0	0	0
BSIs	8/18 (44.4)	7/27 (25.9)	NA/27	5/6 (83.3)	1/1 (100)	5/15 (33.3)	1/4 (25)	3/9 (33.3)	1/2 (50)	3/5 (60)
CRAB	7/15 (46.7)	7/27 (25.9)	NA/27	1/1 (100)	0	5/13 (38.5)	0	3/8 (37.5)	0/1 (0)	3/3 (100)
Septic shock	10/14 (71.4)	NA/30	NA/18	5/6 (83.3)	1/1(100)	NA	3/5 (60)	1/4 (25)	2/5 (40)	NA
30-D Mortality	15/41 (36.6)	16/47 (34)	23/42 (55)	6/10 (60)	1/8 (12.5)	5/18 (27.8)	4/17 (23.5)	3/13 (23.1)	2/5 (40)	10/24 (42)

Where not otherwise defined, results are presented as patients expired/total patients (%). Here **30-D** = thirty-days; **IQR** = interquartile range; **HAP** = hospital-acquired pneumonia; **VAP** = ventilatory-associated pneumonia; **CRAB** = carbapenem-resistant *Acinetobacter baumannii*; **cIAI** = complicated intrabdominal infections; **cUTI** = complicated urinary-tract infections; **SSTI** = soft-skin tissues infections; **CNS** = central nervous system.

## Data Availability

Data are available upon request to the corresponding author.

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
