# Peer review of "Cefiderocol Efficacy in a Real-Life Setting: Single-Centre Retrospective Study"

_antibiotics, 2023, doi:10.3390/antibiotics12040746_

Round 1

Reviewer 1 Report

Authors have presented their centre's results of the use of Cefiderocol in CRGN. Results are from the data of about an year with significant heterogeneity and represents the real world setting use of cefiderocol.

Few suggestions for improving the manuscript writing.

Subgroups analysis based on site of infections and respective duration of treatment is needed for specifying the impact on individual infections. 

Detailed comparison with the CREDIBLE-CR study should be provided.

Materials and methods are grossly incomplete.

A consort diagram should also be provided. 

Adverse events table should also be added.

Dose relationship with survival is not clear in the manuscript. 

Also add data from APEKS-NP study.

Reviewer 2 Report

The paper adds value to literature on the real-world use of cefiderocol, as its a new agent active against MDROs including carbapenem-resistant organisms. There are a few grammatical errors throughout. Some of which I will list below. I would strongly recommend another review by the authors.

The title may need "a" in between in and real-life, so it reads "in a real-life setting".

1) Line 25 does "monocentric" mean single-center retrospective study? If so, I would state "single-center" instead. This can be changed in the title also.

2) Line 44 clarify "non-fermenters" means lactose non-fermenters.

3) Line 55-56 needs review of grammar its difficult to understand the intent of the sentence, does "sustained" mean "caused by"? It sounds like "caused by" CRAB would be better.

4) Lines 97-98 grammar, difficult to understand that sentence. It sounds like it means multiple isolates could have been isolated from patients but only the index isolate (or one isolate) was included and counted in the stats portion. This should be reworded.

5) Were MICs done for any isolates against cefiderocol? I understand testing the agent is difficult with media and iron requirements but any MIC values would be interesting to note if any susceptibility assays were performed.

6)Line 95 "at" may sound better if changed to "in the multivariate model"

7) It may be better to separate table 1 into two different tables. Table 1 can be demographics and table 2 can be on therapy and outcomes. 

8( line 208 may sound better if changed "deceased patient" to "patients that expired"

9) line 218 "not" should be "non" survivors.

10) Line 231 "non-fermenters" again should be "lactose non-fermenters"

11) line 235 "to" is not necessary

12) line 277 "founding" should be "finding"

Overall, the study is very descriptive but adds to the little literature on cefiderocol use in real-world settings.

Round 2

Reviewer 1 Report

I am satisfied with the author's efforts to reply to the comments.